# PPARα Induces the Expression of CAR That Works as a Negative Regulator of PPARα Functions in Mouse Livers

**DOI:** 10.3390/ijms24043953

**Published:** 2023-02-16

**Authors:** Ryota Shizu, Yuta Otsuka, Chizuru Ishii, Kanako Ezaki, Kouichi Yoshinari

**Affiliations:** 1Laboratory of Molecular Toxicology, School of Pharmaceutical Sciences, University of Shizuoka, 52-1 Yada, Suruga-ku, Shizuoka 422-8526, Japan; 2Laboratory of Molecular Toxicology, School of Pharmaceutical Sciences, Tohoku University, 6-3 Aramaki-Aoba, Aoba-ku, Sendai 980-8578, Japan

**Keywords:** CAR, PPARα, nuclear receptor, gene transcription, drug–drug interaction

## Abstract

The nuclear receptor peroxisome proliferator-activated receptor α (PPARα) is a transcription factor that controls the transcription of genes responsible for fatty acid metabolism. We have recently reported a possible drug–drug interaction mechanism via the interaction of PPARα with the xenobiotic nuclear receptor constitutive androstane receptor (CAR). Drug-activated CAR competes with the transcriptional coactivator against PPARα and prevents PPARα-mediated lipid metabolism. In this study, to elucidate the crosstalk between CAR and PPARα, we focused on the influence of PPARα activation on CAR’s gene expression and activation. Male C57BL/6N mice (8–12 weeks old, n = 4) were treated with PPARα and CAR activators (fenofibrate and phenobarbital, respectively), and hepatic mRNA levels were determined using quantitative reverse transcription PCR. Reporter assays using the mouse *Car* promoter were performed in HepG2 cells to determine the PPARα-dependent induction of CAR. CAR KO mice were treated with fenofibrate, and the hepatic mRNA levels of PPARα target genes were determined. Treatment of mice with a PPARα activator increased *Car* mRNA levels as well as genes related to fatty acid metabolism. In reporter assays, PPARα induced the promoter activity of the *Car* gene. Mutation of the putative PPARα-binding motif prevented PPARα-dependent induction of reporter activity. In electrophoresis mobility shift assay, PPARα bound to the DR1 motif of the *Car* promoter. Since CAR has been reported to attenuate PPARα-dependent transcription, CAR was considered a negative feedback protein for PPARα activation. Treatment with fenofibrate induced the mRNA levels of PPARα target genes in *Car*-null mice more than those in wild-type mice, suggesting that CAR functions as a negative feedback factor for PPARα.

## 1. Introduction

Peroxisome proliferator-activated receptor α (PPARα) is a nuclear receptor expressed in the liver that is activated by fatty acids. PPARα controls the transcription of lipid metabolism-related genes, such as *ACOX1*, *HMGCS2*, and *CPT1A* [1]. Fibrates are typical drugs that activate PPARα and induce the expression of PPARα target genes to stimulate lipid metabolism and lower blood triglyceride levels [2]. Depending on fasting or diabetes conditions, PPARα is found to be induced at gene expression levels and activated to induce lipid metabolism [3,4]. Hence, PPARα plays a role in the regulation of energy metabolic homeostasis.

In our recent study, antiepileptic drug-dependent activation of the nuclear receptor constitutive androstane receptor (CAR) prevented the fibrate-activated, PPARα-mediated transcription of genes related to fatty acid metabolism [5]. CAR is a nuclear receptor highly expressed in the liver and plays an important role in xenobiotic-induced gene expression and energy metabolism in the liver [6,7]. Since CAR activation by drugs or food ingredients results in enhanced metabolism or excretion of xeno- or endobiotics, the receptor is one of the major proteins responsible for chemical-dependent toxicity in the liver.

As mentioned above, CAR activation attenuates the fibrate-dependent expression of genes related to fatty acid oxidation and ketogenesis and a decrease in blood triglyceride levels [5]. Mechanistic analyses demonstrated that CAR prevents PPARα by competing with the transcriptional coactivator PGC1α against PPARα [5].

CAR and PPARα are activated in the livers of diabetic or fasted mice [8,9]. Treatment of primary rat hepatocytes with PPARα activators increased the mRNA levels of CAR and its target gene, *Cyp2b1* [10]. These reports suggest a functional interaction between PPARα and CAR in energy metabolism.

In this study, we investigated the detailed molecular mechanisms underlying the interaction between CAR and PPARα in the liver. We found that PPARα positively regulated CAR expression and that CAR attenuated PPARα-dependent gene transcription. Experiments with CAR-knockout mice showed that CAR plays a negative feedback regulatory factor in PPARα-mediated lipid metabolism.

## 2. Results

### 2.1. CAR Inactivates PPARα and PPARα Activates CAR in the Mouse Liver

First, to determine the interaction between CAR and PPARα, mice were treated with the CAR-activating drug phenobarbital (0.03% wt/wt) and PPARα-activating drug fenofibrate (0.03% wt/wt) for one week, and the hepatic mRNA levels of their target genes were determined (Figure 1). Fenofibrate induced the expression of PPARα target genes, such as *Acox1*, *Acat1*, *Hmgcs2*, and *Cyp4a10*. As we previously reported, treatment with phenobarbital attenuated PPARα-dependent gene expression levels of the target genes. Phenobarbital treatment attenuated the fenofibrate-dependent decrease in plasma triglyceride levels. In contrast, *Car* mRNA levels were upregulated by fenofibrate treatment. Additionally, *Cyp2b10* mRNA levels were also induced by fenofibrate treatment. Because PPARα was suggested to be regulated by autoinduction [11], *Ppara* mRNA levels were also upregulated by PPARα activation. In addition, the mRNA levels of *Car* and *Cyp2b10* were induced by treatment with 0.1% wt/wt fenofibrate for 1 week (Figure 1B), which used the same RNA sample shown in Figure 3 of our previous report using 0.1% wt/wt fenofibrate and phenobarbital-treated mice [5]. 

Fasting is known to activate PPARα and induce the expression and activation of CAR. Therefore, we determined the hepatic CAR and PPARα activation levels in fasted mice. Mice were intraperitoneally treated with the CAR ligand TCPOBOP. After 6 h of treatment, mice fasted for 18 h, and their hepatic RNA was collected. As expected, fasting increased the mRNA levels of *Car* and its target *Cyp2b10*, as well as PPARα-targeted *Acox1*, *Hmgcs2*, and *Ppara* mRNA levels (Figure 2). TCPOBOP treatment, which induced *Cyp2b10* mRNA levels, attenuated the fasting-dependent induction of PPARα target genes (Figure 2).

### 2.2. PPARα Induces CAR Expression through Binding to the Novel DR1 Motif in Car Promoter

PPARα induces *Car* mRNA levels in the livers of mice. Between −1970 and +40 bp from the transcription start site of mouse *Car* was cloned into a reporter plasmid to investigate the influence of PPARα activation on the gene transcription of *Car*, and a reporter assay was conducted with an expression of mPPARα in HepG2 cells. As expected, the expression of mPPARα and treatment with its ligand bezafibrate upregulated reporter activity (Figure 3A). The reporter activity upregulated by PPARα was clearly prevented by the expression of murine CAR and treatment with its ligand TCPOBOP (Figure 3A). *Car* mRNA is well known to be mainly controlled by HNF4α [12,13]. When PPARα was co-expressed with HNF4α, additive induction by PPARα was observed (Figure 3B), suggesting that HNF4α and PPARα may differentially regulate CAR expression.

Using JASPER (a web-based transcription element search system; https://jaspar.genereg.net (accessed on 10 January 2019), we identified three DR1 motifs in the mouse *Car* promoter (Figure 3C). To identify the putative PPARα-responsive element among them, we independently introduced mutations into these three DR1 motifs and evaluated the effects of mPPARα on these constructs using reporter assays. The introduction of a mutation into sites 1 and 2 (DR1-1 and DR1-2, respectively) had no influence on the mPPARα-mediated increase in reporter activity. In contrast, a mutation at site 3 (DR1-3) resulted in the complete loss of this increase in reporter activity (Figure 3D). HNF4α-dependent gene transcription was clearly prevented by mutating the DR1-1 motif but not by mutating DR1-2 or DR1-3 (Figure 3D), which is consistent with a previous report showing HNF4α-mediated induction of *Car* expression through this DR1-1 motif [8]. These results suggest that PPARα and HNF4α independently regulate CAR expression. In EMSAs, the mPPARα/RXRα heterodimer bound to the DR1-3 probe but not the DR1-1 and DR1-2 probes, and HNF4α bound to all three probes (Figure 4A). The binding of mPPARα/RXRα with DR1-3 disappeared upon the addition of the non-labeled HMGCS2 probe (Figure 4B). In contrast, the PPARα/RXRα heterodimer did not bind to the mutated DR1-3 probe (Figure 4B). Altogether, these results suggest that PPARα transactivates the mouse *Car* promoter by binding to the novel DR1-3 motif. HNF4α strongly bound to DR1-3 in EMSAs, while it used DR1-1 for the transcription of the *Car* gene.

### 2.3. PPARα Activation Was Enhanced in the Liver of CAR-Knockout Mice

Since CAR attenuated PPARα-dependent gene transcription and PPARα induced the expression of CAR, we hypothesized that CAR acts as a negative regulator of PPARα-dependent gene transcription. Therefore, we treated wild-type and CAR-knockout (CAR-KO) mice with 0.03% wt/wt fenofibrate and determined the hepatic mRNA levels (Figure 5). Fenofibrate increased mRNA levels of PPARα target genes in both wild-type and CAR-KO mice. Compared with wild-type mice, the expression levels of PPARα target genes were increased in CAR-KO mice. *Car* and *Cyp2b10* mRNA levels were increased by fenofibrate treatment in wild-type mice (Appendix A). Induction of *Cyp2b10* mRNA levels by fenofibrate was also observed in CAR-KO mice, suggesting that PPARα may induce the gene expression levels of *Cyp2b10* by activating transcription factors other than CAR. The fenofibrate treatment-dependent decrease in plasma triglyceride levels did not show a further decrease in CAR-KO mice (Appendix A). These results suggest that CAR is activated by PPARα in the liver and may function as a negative regulator of PPARα-dependent gene transcription.

## 3. Discussion

In this study, we investigated the interaction between PPARα and CAR and found that PPARα induced the transcription of the *Car* gene in the mouse liver. Since CAR inhibited PPARα-dependent gene transcription, induced CAR was considered a negative regulator of PPARα-dependent gene transcription. In addition, we determined the molecular mechanism underlying the PPARα-dependent induction of the *Car* gene. This receptor bound to a DR1 motif in the promoter of *Car*. PPARα has been reported to upregulate hepatic mRNA levels of CAR and its target *Cyp2b* genes [10,14]. Our results may support these results and help define the role of CAR in energy metabolism.

PPARα directly bound to the 5′-upstream promoter of mouse *Car*. The expression levels and transcriptional activity of PPARα are upregulated under fasting or diabetic conditions, and PPARα thus controls energy metabolic homeostasis by inducing lipid metabolism [15,16]. CAR is also upregulated under these conditions, and CAR activation in obese mice, such as *ob*/*ob* mice, has been reported to improve insulin resistance and decrease blood glucose levels [17,18]. Moreover, PPARα knockout was reported to rescue obese mice from insulin resistance [19,20]. HNF4α acts as a transcription factor for *Car* expression and induces CAR levels under fasting conditions [8]. Considering our results, fasting- or diabetes-dependent induction of CAR is regulated by PPARα as well as HNF4α. Interestingly, the DR1 motif for PPARα is different from that used for HNF4α-dependent *Car* transcription. Although our experiments on mechanistic analysis are limited to a reporter assay and EMSA with in vitro translated protein, and more extensive studies, such as, for example, those using a ChIP assay or ChIP-seq assay, are required, it is suggested that CAR is involved in hepatic energy metabolism and plays a role in energy homeostasis by crosstalking with the PPARα-mediated signaling. 

Since CAR was found to attenuate PPARα-dependent gene expression, it may work as a negative feedback factor for PPARα activation. Fenofibrate treatment in CAR-KO mice showed that the induction of PPARα target gene levels was higher than that in wild-type mice. These results support the strategy of a negative feedback loop based on CAR activation.

In this study, we found that PPARα directly bound to the DR1 motif in the promoter region of the *Car* gene. However, the molecular mechanism by which PPARα activates CAR (i.e., induces the nuclear accumulation of CAR) remains unclear. Streptozotocin-treated type 1 diabetes model mice showed induced expression of the CAR target gene, *Cyp2b10*, following an increase in blood glucose levels, and insulin treatment clearly attenuated these inductions [9]. Several reports have suggested that hepatic adenosine monophosphate-activated protein kinase (AMPK), a serine/threonine protein kinase, is activated under low-energy conditions as an energy and nutrition sensor [9,21,22]. Since it has been reported that the nuclear CAR level is elevated by phosphorylated AMPK, diabetes-dependent CAR activation may be caused by the activation of AMPK [23,24]. On the other hand, the inducible nuclear receptor coactivator PGC1α is upregulated by several transcription factors (including PPARs) in several energy conditions, such as fasting [25,26] and is important for fasting-dependent gene transcription, including *Car* [8,26]. PGC1α strongly activates CAR-dependent transcription [5], whereas CAR suppresses PGC1α-related gene transcription and expression of hepatic gluconeogenic genes [27]. These proteins may play a role in the CAR-mediated induction of *Cyp2b10* under low-energy conditions.

Recently, we reported that the nuclear receptor PXR also attenuated the PPARα-dependent transcription of its target genes [28]. It is reported that PXR is also induced in the livers of mice under fasting conditions [29]. In addition, our preliminary experiment suggested that fenofibrate treatment-mediated PPARα activation in the mouse liver increased mRNA levels of PXR as well as CAR. Moreover, *Cyp2b10*, which is upregulated by both CAR and PXR, was induced by fenofibrate treatment in the livers of CAR-KO mice (Appendix A). It is thus possible that PXR expression is also induced by PPARα, and it acts as a negative feedback regulator of PPARα, as does CAR.

## 4. Materials and Methods

### 4.1. Materials

1,4-Bis [2-(3,5-dichloropyridyloxy)] benzene (TCPOBOP), bezafibrate, and fenofibrate were purchased from Sigma-Aldrich (St. Louis, MO, USA). Sodium phenobarbital was purchased from FUJIFILM Wako Pure Chemicals (Osaka, Japan). Oligonucleotides were synthesized by Macrogen (Seoul, Korea). All other reagents were obtained from FUJIFILM Wako Pure Chemical or Sigma-Aldrich unless otherwise indicated.

### 4.2. Animal Experiment

Male C57BL/6N mice (8–12 weeks old) were maintained under a 12 h light/dark cycle and fed a conventional laboratory diet and water. Mice were fed a purified diet (CE-2; Clea Japan, Tokyo, Japan) supplemented with or without fenofibrate (0.03 or 0.1% wt/wt) and phenobarbital (0.03 or 0.1% wt/wt) for a week. Mice were intraperitoneally treated with TCPBOP (3 mg/kg dissolved in corn oil). Six hours after treatment, the mice fasted for 18 h. Livers and plasma were collected and subjected to experiments. Plasma triglyceride levels were determined using a Triglyceride E Test WAKO kit (Fujifilm Wako Pure Chemicals). All animal experiments were approved by the committees for animal experiments at Tohoku University and the University of Shizuoka and conducted in accordance with the guidelines for animal experiments at Tohoku University and the University of Shizuoka.

### 4.3. Plasmid Preparation

The mCAR and mPPARα expression plasmids (mCAR pTargeT and mPPARα pTarget) [5] and the mHNF4α expression plasmid (mHNF4α pTargeT) [30] were previously prepared. The phRL-TK-, pGL4.31-, pGL4.10-, and PGC1α-expressing pFN21A plasmids were purchased from Promega (Madison, WI, USA). mCar-2k pGL4.10 were constructed with the insertion of −1970 to +66 of *mCar* promoter into XhoI and EcoRV sites of pGL4.10. Mutated plasmids were produced by PCR using a KOD Plus mutagenesis kit (TOYOBO, Otsu, Japan) with specific primer sets. The sequences of the point mutation of mCar-2k DR1-1mut, DR1-2mut, and DR1-3mut are shown in Figure 3C.

### 4.4. Cell Culture

HepG2 cells (RIKEN BioResource Center, Tsukuba, Japan) were cultured in a Dulbecco’s modified Eagle’s medium (DMEM; FUJIFILM Wako Pure Chemical) supplemented with heat-inactivated 10% fetal bovine serum (GE Healthcare, Buckinghamshire, UK), nonessential amino acids (Thermo Fisher Scientific, Waltham, MA, USA), and antibiotic-antimycotic (Thermo Fisher Scientific). Cells were seeded in 96-well plates (BD Biosciences, Heidelberg, Germany) at 1 × 10^4^ cells/well. Twenty-four hours after the seeding, plasmid transfection was conducted using a Lipofectamine 3000 (Invitrogen, Carlsbad, CA, USA) following the instructions.

### 4.5. RT-qPCR

Total RNA was isolated using Sepasol RNA I (Nacalai Tesque, Kyoto, Japan). RNA was reverse-transcribed by a High-Capacity cDNA Reverse Transcription Kit (Thermo Fisher Scientific). Quantitative reverse transcription-PCR (RT-qPCR) was carried out using GoTaq qPCR Master Mix (Promega) and StepOnePulsTM Real-Time PCR Systems (Applied Biosystems) with specific primer sets for each target gene, as previously reported [5]. Target gene mRNA levels were normalized to *Actb* mRNA levels.

### 4.6. Reporter Assays

Twenty-four hours after seeding, cells were co-transfected with reporter gene plasmid, expression plasmid, and *Renilla* luciferase-expressing plasmid using Lipofectamine 3000 (Invitrogen) and treated with vehicle (0.1% or 0.2% dimethyl sulfoxide, DMSO) or drugs in serum-free DMEM for 24 h. Cells were lysed with Passive Lysis Buffer (Promega) and reporter activity was measured using Dual-Luciferase Reporter Assay System (Promega) following the manufacturer’s instructions. Firefly luciferase luminescence was normalized to *Renilla* luciferase luminescence.

### 4.7. Electrophoresis Mobility Shift Assay (EMSA)

EMSAs were performed as previously described [5]. Briefly, mPPARα, hRXRα, and mHNF4α were synthesized with a pTNT plasmid using the TnT SP6 Quick-Coupled Transcription/Translation System (Promega). ^32^P-labeled probes for the mouse *Car* promoter DR1-1, 5′-ACCCAGGTCTTTGCCCTGGGT-3′; DR1-2, 5′-ACACTGTCCTCTGATCTCTGT-3′; and DR1-3, 5′-ATAAAGGTCAGAGAACAACTT-3′, and the *Hmgcs2* promoter, 5′-AGTGAGCCCTTTGACCCAGTT-3′ were incubated with the in vitro synthesized proteins, and the reaction mixture was subjected to electrophoresis. The underlined probe sequences represent nuclear receptor-binding motifs.

### 4.8. Statistical Analysis

Statistical analyses were performed using GraphPad Prism 9 (Ver. 9.5.1, GraphPad Software, San Diego, CA, USA). The significance of differences was assessed using the Student’s *t*-test for the comparison of data from two groups and one-way ANOVA followed by Bonferroni’s correction or Dunnett’s test for the comparison of multiple group data, based on the experimental design. The *p*-values less than 0.05 were regarded as statistically significant, and asterisks indicate statistical significance. The values were not used for testing the experimental hypotheses but were indicated to understand the differences between the compared groups. All experiments were repeated at least twice to confirm reproducibility. Sample sizes were specified before conducting the experiments, and the number of experiments to check the reproducibility was determined after the initial results were obtained.

## 5. Conclusions

We have found that PPARα induces CAR activation, and activated CAR prevents PPARα-dependent gene transcription. We have also revealed a molecular mechanism for the PPARα-mediated induction of CAR expression. Since CAR and PPARα are activated by a number of pharmaceutical drugs, herbs, and food components, the activation of these receptors might cause drug–drug interactions and affect the induction of drug-metabolizing enzymes or fatty acid oxidation and ketogenesis.

## Figures and Tables

**Figure 1 ijms-24-03953-f001:**
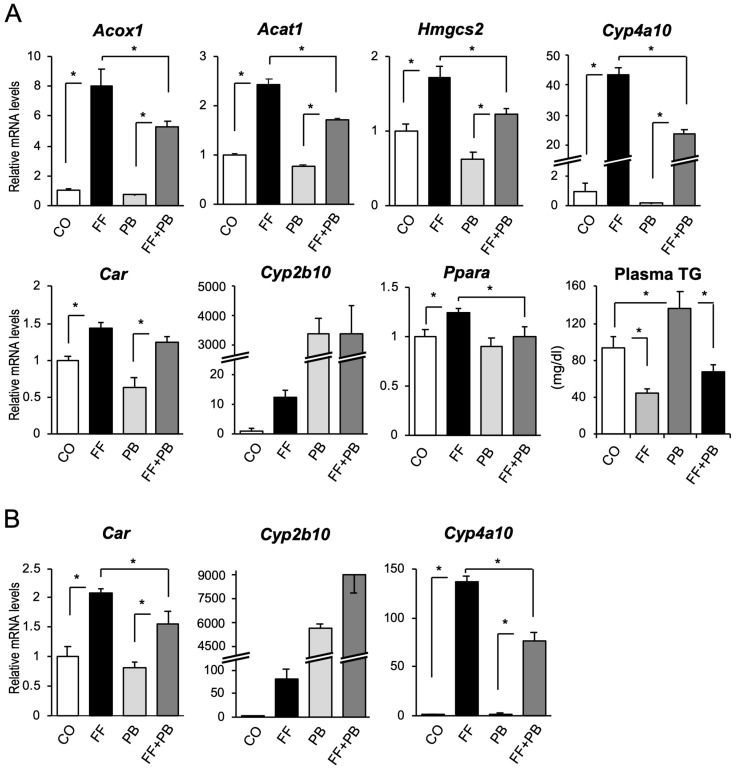
The influence of PPAR- and CAR-activator treatment on the hepatic mRNA levels of PPARα and CAR target genes in mice. (**A**) Mice were fed a CE-2 chow containing fenofibrate (FF) and/or phenobarbital (PB) at 0.03% wt/wt for one week. CO, control. Hepatic total RNA from the suggested genes was subjected to RT-qPCR, and plasma TG was determined. (**B**) cDNA samples of Figure 3 from our previous report [5] were used to determine the mRNA levels of *Car*, *Cyp2b10*, and *Cyp4a10*. Data shown are the mean ± S.D. (n = 4); statistical analyses were performed with comparisons between the groups via Bonferroni’s correction (* *p* < 0.05).

**Figure 2 ijms-24-03953-f002:**
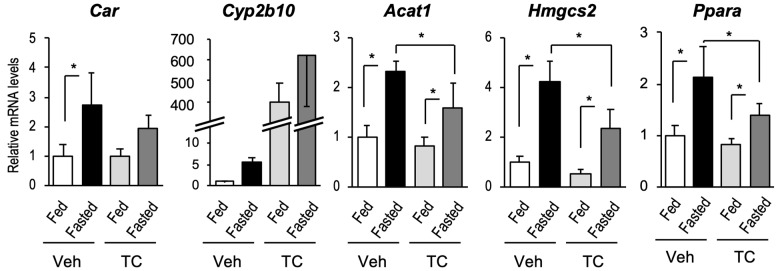
The influence of fasting on the expression of PPAR and CAR target genes. Mice were intraperitoneally treated with TCPOBOP (TC, 3 mg/kg) or vehicle (corn oil, Veh). Six hours after treatment, the mice fasted for 18 h. Total hepatic RNA from the suggested genes was subjected to RT-qPCR. Data are shown as the mean ± S.D. (n = 4); statistical analyses were performed with comparisons between the groups via Bonferroni’s correction (* *p* < 0.05).

**Figure 3 ijms-24-03953-f003:**
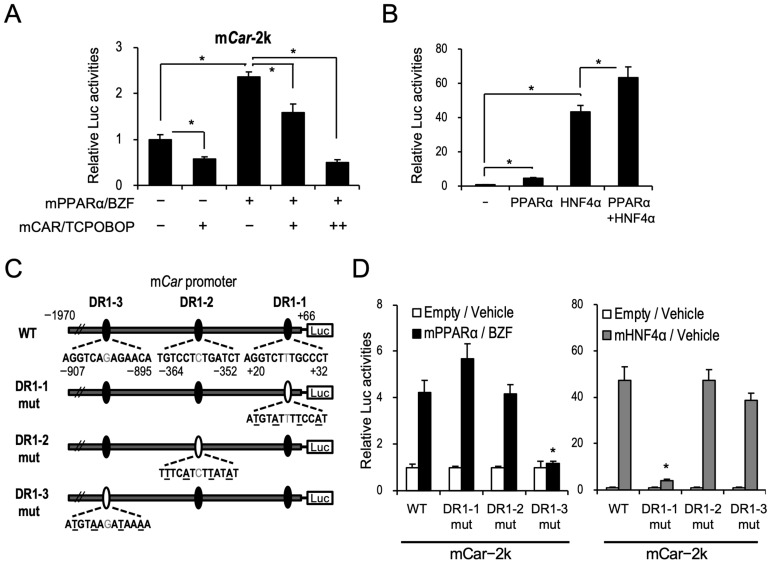
(**A**) HepG2 cells were transfected with the mCar-2k pGL4.10 reporter plasmid containing −1970 to +66 of mCar promoter (250 ng) and phRL-SV40 (5 ng), and an expression plasmid for mPPARα (25 ng) and mCAR (+, 2.5 ng; ++, 25 ng). Twelve hours after transfection, the cells were treated with vehicle (DMSO), TCPOBOP (250 nM), and/or BZF (100 µM) for 24 h. Firefly luciferase activities were normalized with *Renilla* luciferase activities. (**B**) HepG2 cells were transfected with the mCar-2k pGL4.10 reporter plasmid (250 ng) and phRL-SV40 (5 ng), and an expression plasmid for mPPARα (25 ng) and mHNF4α (25 ng), and reporter activity was determined. Statistical analyses were performed with comparisons between the groups via Bonferroni’s correction (* *p* < 0.05). (**C**) Putative PPARα-binding motifs of the mouse Car promoter were proposed, which were three DR1 motifs named DR1-1, DR1-2, and DR1-3. (**D**) Reporter assays were carried out with the wild-type and mutated reporter constructs containing point mutation of each DR1 motif, as shown in (**C**). Data are shown as the mean ± S.D. (n = 4); * *p* < 0.05 (Dunnett’s test versus the corresponding mCAR-2k-WT expressed groups).

**Figure 4 ijms-24-03953-f004:**
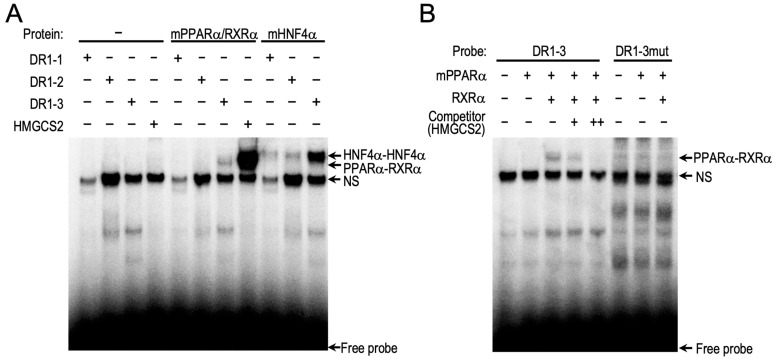
Binding of PPARα and HNF4α on the promoter of mouse *Car*. EMSA was performed as described in the Materials and Methods section using 32P-labeled probes and in vitro synthesized nuclear receptor proteins. (**A**) Synthesized nuclear receptor proteins were incubated with each probe. The probe for the DR1 motif in the *Hmgcs2* promoter was used as a positive control. (**B**) The 32P-unlabeled probes of the DR1 motif in Hmgcs2 were co-incubated as competitors.

**Figure 5 ijms-24-03953-f005:**
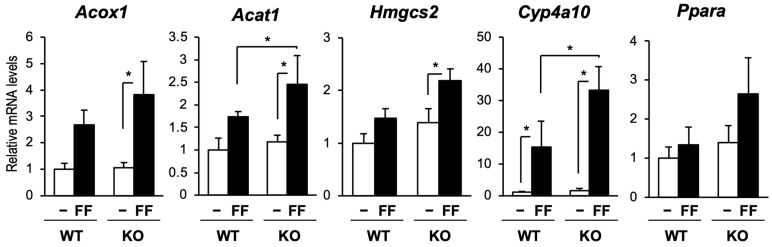
The influence of CAR knockout on PPARα-dependent gene expression. Wild-type (WT) and CAR-knockout (KO) mice were fed CE-2 chow containing fenofibrate (FF) at 0.03% wt/wt for 1 week. Hepatic RNA was extracted and subjected to RT-qPCR, as suggested. Data are shown as the mean ± S.D. (n = 4); statistical analyses were performed with comparisons between the groups via Bonferroni’s correction (* *p* < 0.05).

## Data Availability

The datasets generated and analyzed in this study are included within the manuscript and Appendix A and can be obtained from the corresponding authors upon reasonable request.

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
