# Peer review of "PPARα Induces the Expression of CAR That Works as a Negative Regulator of PPARα Functions in Mouse Livers"

_ijms, 2023, doi:10.3390/ijms24043953_

Round 1
Reviewer 1 Report
Shizu and collaborators have studied the connection between PPARalpha and CAR pathways in the liver of mice. To do so, they have fed WT mice with either FF, PB or a combo of both (FF+PB) and have evaluated hepatic mRNA levels of PPARalpha and CAR target genes.CAR-KO mice were also fed with fenofibrate and the induction of PPARalpha target genes levels was also quantified vs WT mice. Together, it was found that CAR would attenuate the PPARalpha capacity to mediate upregulaton of its own target genes. CAR was also identified as a direct PPARalpha target gene in liver of mice since a functionnal DR1 (PPRE) was clearly identified within the close CAR promoter sequence.
Together, this is an interesting study that should be improved following the different points listed below :
Major points :
Point 1 : Pparalpha mRNA expression levels should be presented in Figure 1A, Figure 2 and Figure 5.
Point 2 : In an effort to strenghten the EMSA data (Fig 4A and 4B) dealing with the heterodimer PPARalpha/RXRalpha, DNA ChiP would be a real asset here, given that the sole DNA probe incubation alone gives here a strong background signal (see arrow NS). Moreover, EMSA becomes a method from another age that works here with « in vitro translated protein » which may be a bit far from the real physiological situation with post-modified (phosphorylation, glycosylation, etc etc) proteins.
Point 3 : DR1-3 is a potent binding site for the HNF4alpha/HNF4alpha homodimer (Fig. 4A). Yet, when mutated DR1-3 is used for transactivation (Fig. 3D, right panel) with mCar-2K promoter sequence, it turns out that transactivation is somewhat similar to that of the WT sequence. How do you explain it ? Did you check for the mutation ?
Point 4 : Why not testing the DR1-3 DNA mutated probe in EMSA together with the heterodimer PPARalpha/RXRalpha proteins ? (fig. 4B). If you get the results, please show them.
Point 5 : Statistical significance has to be shown when necessary (Fig.3 A, B and D.)
Point 6 : Lane 201-202 : It turns out that upon reading this sentence, PPARalpa may physically interacts with CAR. Yet, you do not have any evidence here, in the present study, that supports this statement. Please, change the sentence saying that for instance « both CAR and PPARalpha pathways might be inter-connected ».
Point 7 : Mice and rats: Gene symbols are italicized, with only the first letter in upper-case (e.g., Gfap). Protein symbols are not italicized, and all letters are in upper-case (e.g., GFAP). Please, correct throughout the MS accordingly.
Point 8 : It is absolutely mandatory to provide more informations about the RT-qPCR procedure. Which RT kit, total RNA amount/quantity that has been retro-transcribed, which DNA Taq Pol enzyme was used, SYBR Green or TaqMan Probes ?, primer sequences, program step, etc, etc.
Minor points :
- Lanes 72 and 147 and also Figure 5 : Change « transcription » by « gene expression levels ». Indeed, RT-qPCR is not indicative of gene transcription, rather a number of transcripts at a particular time point (that reflects both gene transcription and mRNA degradation depending on the half-life of the target mRNA).
- Lane 127 : consistent with a previous report,
- Lane 127 : Hnf4a (a in symbol)
- 4.3 : Plasmid preparation
Please, provide the readers with the adequate informations regarding the point mutations (give sequence) of the CAR Response Element (mandatory) used in Fig. 3D.
- Lane 161 : Putative PPARα binding motifS
- Please, recall what is « ppm » when you are talking about fenofibrate or phenobarbital. Would rather prefer the same nomenclature as that present lanes 250 and 251.
- Lane 188 : binds to a DR1 instead of THE DR motif
- 4.2 Lane 252 : Please, clearly indicate the dose of TCPBOP used (see lane 110… mg/kg) and
- 4.4 Which liposomes did you use to perform transfection experiments.
- Lane 110 : Please, clearly indicate how much TCPOBOP was used (mg/kg)
- Lane 111 : RTqPCR instead of RT-PCR and throughout the text.
Author Response
Major points :
Point 1 : Pparalpha mRNA expression levels should be presented in Figure 1A, Figure 2 and Figure 5.
Our response: We have added Pparalpha mRNA levels into Figure 1A, Figure 2 and Figure 5. Following this, we have added the following sentence in the results section, “Since PPARα was suggested to be regulated by autoinduction [11], Ppara mRNA levels were also upregulated by PPARα activation.” In addition, citation 11 has been added. (Lines 74-76, Page 2.)
Point 2 : In an effort to strenghten the EMSA data (Fig 4A and 4B) dealing with the heterodimer PPARalpha/RXRalpha, DNA ChiP would be a real asset here, given that the sole DNA probe incubation alone gives here a strong background signal (see arrow NS). Moreover, EMSA becomes a method from another age that works here with « in vitro translated protein » which may be a bit far from the real physiological situation with post-modified (phosphorylation, glycosylation, etc etc) proteins.
Our response: We agree with the reviewer’s suggestion. EMSA indicates just a DNA binding capacity of proteins and this binding information is not enough for understanding transcription as described in your point 3. Also, more than 90% of in vitro translated proteins are folded inaccurately and may form an inclusion body. Despite this, EMSA is an established method to determine at least the binding capacity of accurately folded proteins with specific DNA oligomers. What we wanted to demonstrate in Fig. 4 was the information on the DNA binding capability of PPARalpha/RXRalpha and HNF4alpha. So, we think that EMSA is enough for this purpose. Yet, we have added the following sentence, “Although our experiments on mechanistic analysis are limited to reporter assay and EMSA with in vitro translated protein and more extensive studies, for example, those using ChIP assay or ChIP-seq assay are required,…” in lines 201-203, page 7.
Point 3 : DR1-3 is a potent binding site for the HNF4alpha/HNF4alpha homodimer (Fig. 4A). Yet, when mutated DR1-3 is used for transactivation (Fig. 3D, right panel) with mCar-2K promoter sequence, it turns out that transactivation is somewhat similar to that of the WT sequence. How do you explain it? Did you check for the mutation?
Our response: We think binding on the promoter region is not enough for gene transcription. DR1-3 showed a strong binding affinity with HNF4alpha but the motif is not important for gene transcription. Considering the reviewer's comment, we have added the following sentence in the results section; “HNF4α strongly bound to DR1-3 in EMSAs while it used DR1-1 for the transcription of Car gene.” in lines 135-136, page 4. The DR1-3-mutated plasmid contains the DR1-1 motif, and thus HNF4alpha homodimer should bind to DR1-1 and induce the reporter activity by binding to the DR1-1 motif.
Point 4 : Why not testing the DR1-3 DNA mutated probe in EMSA together with the heterodimer PPARalpha/RXRalpha proteins ? (fig. 4B). If you get the results, please show them.
Our response: We had obtained the data of DR1-3 mutated probe in EMSA but in the original manuscript, we did not include the data as a figure with the description “PPARα/RXRα heterodimer did not bind to the mutated DR1-3 probe (data not shown)”. Following the reviewer’s comment, we have added the data in Fig. 4B and removed “(data not shown)” from the manuscript. (Line 133, page 4.)
Point 5 : Statistical significance has to be shown when necessary (Fig.3 A, B and D.)
Our response: These data are from cultured cells and the sample size was n = 4. Therefore, we think a statistical comparison of these groups is not meaningful. Therefore, we have not added the asterisk and the statement for statistical analysis in Fig. 3.
Point 6 : Lane 201-202 : It turns out that upon reading this sentence, PPARalpa may physically interacts with CAR. Yet, you do not have any evidence here, in the present study, that supports this statement. Please, change the sentence saying that for instance « both CAR and PPARalpha pathways might be inter-connected ».
Our response: We have changed the sentence to “it is suggested that CAR is involved in hepatic energy metabolism and plays a role in energy homeostasis by cross-talking with the PPARα-mediated signaling.”, from “These reports suggest that CAR may be involved in hepatic energy metabolism and plays a role in energy homeostasis by interacting with PPARα.” (Lines 203-205, page 7)
Point 7 : Mice and rats: Gene symbols are italicized, with only the first letter in upper-case (e.g., Gfap). Protein symbols are not italicized, and all letters are in upper-case (e.g., GFAP). Please, correct throughout the MS accordingly.
Our response: We have checked and corrected them as suggested throughout the manuscript.
Point 8 : It is absolutely mandatory to provide more informations about the RT-qPCR procedure. Which RT kit, total RNA amount/quantity that has been retro-transcribed, which DNA Taq Pol enzyme was used, SYBR Green or TaqMan Probes ?, primer sequences, program step, etc, etc.
Our response: We have added the information. (Lines 275-280, page 8)
Minor points :
- Lanes 72 and 147 and also Figure 5 : Change « transcription » by « gene expression levels ». Indeed, RT-qPCR is not indicative of gene transcription, rather a number of transcripts at a particular time point (that reflects both gene transcription and mRNA degradation depending on the half-life of the target mRNA).
Our response: We have corrected them as suggested.
- Lane 127 : consistent with a previous report,
Our response: We have corrected it as suggested.
- Lane 127 : Hnf4a (a in symbol)
Our response: we have corrected it as suggested.
- 4.3 : Plasmid preparation
Please, provide the readers with the adequate informations regarding the point mutations (give sequence) of the CAR Response Element (mandatory) used in Fig. 3D.
Our response: We have added the information of the sequence in Fig. 3C and modified the legend to Fig. 3 and described it in “4.3: Plasmid preparation” as follows: “The sequences of the point mutation of mCar-2k DR1-1mut, DR1-2mut, and DR1-3mut are shown in Fig. 3C.”
- Lane 161 : Putative PPARα binding motifS
Our response: We have corrected it as suggested.
- Please, recall what is « ppm » when you are talking about fenofibrate or phenobarbital. Would rather prefer the same nomenclature as that present lanes 250 and 251.
Our response: As the reviewer suggested, “ppm” is not an SI unit and some readers may be confused. We have thus changed “ppm” to “% wt/wt”
- Lane 188 : binds to a DR1 instead of THE DR motif
Our response: We have corrected it as suggested.
- 4.2 Lane 252 : Please, clearly indicate the dose of TCPBOP used (see lane 110… mg/kg) and
Our response: We have added the dose of TCPOBOP.
- 4.4 Which liposomes did you use to perform transfection experiments.
Our response: We used Lipofectamine 3000, which was written in the section of “4.6 Reporter Assay” in the manuscript. But following the reviewer's comment, we have added the information also in “4.4 Cell Culture section”
- Lane 110 : Please, clearly indicate how much TCPOBOP was used (mg/kg)
Our response: We have corrected it to (3 mg/kg)
- Lane 111 : RTqPCR instead of RT-PCR and throughout the text.
Our response: We have used the term “quantitative RT-PCR” throughout the text in the original manuscript. But following the reviewer’s comment, we have changed them to “RT-qPCR”.
Reviewer 2 Report
Dear Authors, the article is very interesting, but it requires many corrections and explanations, which I attach below:
1. In the abstract: the number of mice and whether they were healthy were not given? obese? (there must be information about the mouse population under study). What is the information that we have recently reported a possible drug-drug interaction mechanism via the interaction of PPARα with the liver-highly expressed xenobiotic nuclear receptor constitutive androstane receptor (CAR)?
2. the introduction should be followed by a description of the materials and methods, not the results. The conclusions subchapter should be separated after discussion.
3. in intoduction line 58-53 - this sentence should be in the results/discussion, not in the introduction.
4. in the materials and methods section, there should be information about what was done in the study (step by step) - some of this information is included in the results.
Taking into account these comments, the work requires further evaluation.
Author Response
1. In the abstract: the number of mice and whether they were healthy were not given? obese? (there must be information about the mouse population under study). What is the information that we have recently reported a possible drug-drug interaction mechanism via the interaction of PPARα with the liver-highly expressed xenobiotic nuclear receptor constitutive androstane receptor (CAR)?
Our response: Following the comment, we have added the information on the mice (Line 17, Page 1). The information that we have recently reported a possible drug-drug interaction mechanism via the interaction of PPARα with the liver-highly expressed xenobiotic nuclear receptor constitutive androstane receptor (CAR) is described in the next sentence as follows: “Drug-activated CAR competes with the transcriptional coactivator against PPARα and prevents PPARα-mediated lipid metabolism.” The period after “(CAR)” has been changed to the colon. (Line 14, page 1)
2. the introduction should be followed by a description of the materials and methods, not the results. The conclusions subchapter should be separated after discussion.
Our response: According to the instruction of IJMS, the Materials and Methods should be placed after the discussion section. Thus, we have not changed the order.
According to the reviewer’s comment, we have separated the conclusion (as section 5) from the discussion following the instruction of IJMS.
3. in introduction line 58-53 - this sentence should be in the results/discussion, not in the introduction.
Our response: We had added this short paragraph on lines 57-61, according to the instruction, which says “Introduction section needs briefly mention the main aim of the work and highlight the main conclusions”. But, considering the reviewer’s comment, we have made this paragraph shorter.
4. in the materials and methods section, there should be information about what was done in the study (step by step) - some of this information is included in the results.
Our response: This point was also commented on by Reviewer 1. We have thus added more detailed information throughout the materials and methods section.
Round 2
Reviewer 1 Report
Dear authors,
Thank you for having provided a revised version of your MS. I'm almost fully satisfied with the comments and the novel experiments/data you have brought here.
Yet, concerning figures 3A, 3B, 3D: I still have concerns dealing with statistical analysis. You claimed that n=4; What do you mean? n=4 wells and that's all ? or n=4 different batches of cultured cells.
This is completely different. If 4 different batches of cultured cells have been used (with reasonnably 3 wells/condition) , you should be able to provide the readers some statistical analyses.
In contrast,if you did experiment with n=4 wells of a single cell culture, it means that you should repeat the experiment until you get data from 3 different batches of cultured HepG2 cells.
So please, clarify this point.
Author Response
Reviewer1
Dear authors,
Thank you for having provided a revised version of your MS. I'm almost fully satisfied with the comments and the novel experiments/data you have brought here.
Yet, concerning figures 3A, 3B, 3D: I still have concerns dealing with statistical analysis. You claimed that n=4; What do you mean? n=4 wells and that's all ? or n=4 different batches of cultured cells.
This is completely different. If 4 different batches of cultured cells have been used (with reasonnably 3 wells/condition), you should be able to provide the readers some statistical analyses.
In contrast, if you did experiment with n=4 wells of a single cell culture, it means that you should repeat the experiment until you get data from 3 different batches of cultured HepG2 cells.
So please, clarify this point.
Our response: Thank you for the valuable comment. We conducted the experiment with n=4 well of cultured cells and repeated the same experiment several times to confirm reproducibility, which was written in 4.8.Statistical analysis section. Following the comment, we performed statistical analysis on the data in Figure 3A, 3B and 3D and showed the results in Figures 3 with asterisks. We also added the information of statical analysis on the legend of Fig. 3.
Reviewer 2 Report
Dear Authors. The changes introduced are fully satisfying for me. I recommend the article for publication.
Author Response
Dear Authors. The changes introduced are fully satisfying for me. I recommend the article for publication.
Our response:
Thank you for reviewing our manuscript and valuable suggestions.